# β-catenin-promoted cholesterol metabolism protects against cellular senescence in naked mole-rat cells

Woei-Yaw Chee[1], Yuriko Kurahashi[1], Junhyeong Kim[1], Kyoko Miura [2], Daisuke Okuzaki [3,4], Tohru Ishitani [5], Kentaro Kajiwara [1], Shigeyuki Nada[1], Hideyuki Okano [6] & Masato Okada [1✉]

The naked mole-rat (NMR; *Heterocephalus glaber*) exhibits cancer resistance and an exceptionally long lifespan of approximately 30 years, but the mechanism(s) underlying increased longevity in NMRs remains unclear. In the present study, we report unique mechanisms underlying cholesterol metabolism in NMR cells, which may be responsible for their anti-senescent properties. NMR fibroblasts expressed β-catenin abundantly; this high expression was linked to increased accumulation of cholesterol-enriched lipid droplets. Ablation of β-catenin or inhibition of cholesterol synthesis abolished lipid droplet formation and induced senescence-like phenotypes accompanied by increased oxidative stress. β-catenin ablation downregulated apolipoprotein F and the LXR/RXR pathway, which are involved in cholesterol transport and biogenesis. Apolipoprotein F ablation also suppressed lipid droplet accumulation and promoted cellular senescence, indicating that apolipoprotein F mediates β-catenin signaling in NMR cells. Thus, we suggest that β-catenin in NMRs functions to offset senescence by regulating cholesterol metabolism, which may contribute to increased longevity in NMRs.

[1] Department of Oncogene Research, Research for Microbial Disease, Osaka University, Suita, Osaka, Japan. [2] Department of Aging and Longevity Research, Faculty of Life Sciences, Kumamoto University, Kumamoto, Japan. [3] Genome Information Research Center, Research Institute for Microbial Diseases, Osaka University, Osaka, Japan. [4] Human Immunology Lab, WPI Immunology Frontier Research Center, Institute for Open and Transdisciplinary Research Initiatives, Osaka University, Osaka, Japan. [5] Department of Homeostatic Regulation, Research for Microbial Disease, Osaka University, Suita, Osaka, Japan. [6] Department of Physiology, Keio University School of Medicine, Shinjuku-ku, Tokyo, Japan. ✉email: okadam@biken.osaka-u.ac.jp

Naked mole-rats (NMRs; *Heterocephalus glaber*) are known for their exceptional longevity and remarkable resistance to cancer[1,2]; indeed, only two cases of cancer reported in captive NMRs were reported after multi-year observation of large colonies[3]. In addition, NMRs are strictly subterranean mammals that live in low-oxygen environments[4]; therefore, they exhibit marked resistance to hypoxia[5]. Interestingly, NMRs can survive in oxygen-deprived (anoxia) conditions for 18 min without noticeable injury[6]. Despite accumulating considerable levels of oxidative damage and protein carbonylation under anoxic conditions, NMRs appear to be resilient to oxidative stress and mitochondrial injury, which is strikingly accompanied by a slower aging rate and increased longevity[7–9]. In addition, NMRs display negligible senescence accompanied by high fecundity, and most importantly, remain healthy and are resistant to age-related diseases[10].

These attributes mean that the NMR has been utilized increasingly as an animal model for human aging and cancer research. Several cancer-resistant models have been described in this species. For example, NMR fibroblasts exhibit extreme sensitivity to contact inhibition in tissue culture, which is a potential anticancer mechanism regulated by INK4[11]. An additional study demonstrated that hyaluronan, a high molecular mass polysaccharide of the extracellular matrix, triggers early contact inhibition[2]. Furthermore, treatment with a combination of oncoproteins that trigger tumor formation in mouse cells does not cause malignant transformation of NMR cells[12], corroborating evidence suggesting that the NMR is resistant to both spontaneous cancer development and experimentally-induced tumorigenesis[13]. Furthermore, Miyawaki et al. reported that NMR-derived induced pluripotent stem cells are also tumor resistant[14].

To identify the mechanisms of longevity and cancer resistance in NMRs, we conducted comparative analyses of oncogenic signaling between NMR skin/lung fibroblasts (NSFs/NLFs), mouse skin fibroblasts (MSFs), and NIH 3T3 cells. We found that NMR cells showed altered Wnt/β-catenin signaling. Basal β-catenin expression was significantly higher in NMR cells than in mouse cells. In addition, *β-catenin* knockdown in NSFs induced senescence-like phenotypic changes. Meanwhile, we observed abundant lipid droplets with high levels of cholesterol in NMR cells. Because both *β-catenin* knockdown and cholesterol synthesis inhibition abolished lipid droplet formation and promoted senescence-like phenotypes, we investigated the functional link between β-catenin signaling, cholesterol metabolism, and cellular senescence. Our findings suggest that β-catenin-promoted cholesterol metabolism is crucial for protecting NMR cells from cellular senescence.

## Results

**Altered Wnt/β-catenin signaling in NMR cells**. First, we compared expression levels of various components in oncogenic signaling pathways in NSFs/NLFs, MSFs, and NIH 3T3 cells. We found that β-catenin, a critical transcriptional regulator of the Wnt/β-catenin pathway, was upregulated markedly in NMR cells compared with mouse cells, yet the inactive phosphorylated form of β-catenin was undetectable (Fig. 1a). Alignment of β-catenin amino acid sequences from mouse/human and NMR revealed that β-catenin was highly conserved between species; only one amino acid residue differed between the two (Supplementary Fig. S1). Thus, we analyzed β-catenin levels and activity in NMR cells using an antibody cross-reactive with multiple species and an assay method developed for human and mouse β-catenin. To investigate whether the culture conditions affected the abundance of β-catenin in NMR and mouse cells, we cultured the cell lines at different serum concentrations, oxygen concentrations, and temperatures, and compared β-catenin levels. The results showed that β-catenin expression levels in NSFs were considerably higher than those in MSFs under any conditions, although they were relatively low at high temperature and hypoxia conditions, potentially due to low cell viability of NMR cells under such stressful conditions (Supplementary Fig. S2).

Furthermore, expression of Axin1, a negative regulator of Wnt signaling, was downregulated markedly in NMR cells (Fig. 1a). Immunofluorescence analysis revealed that β-catenin was distributed widely in both the cytoplasm and nucleus of NSFs (Fig. 1b). In addition, a TCF/LEF-dependent TOPFLASH reporter assay revealed that transcriptional activity of β-catenin was significantly higher in NMR cells than in mouse (Fig. 1c). These findings implied that both translocation of β-catenin to the nucleus and β-catenin signaling were constitutively active in NMR cells. However, β-catenin abundance was not affected by the treatment with IWP2, an inhibitor that targets the membrane-bound O-acyltransferase porcupine to prevent Wnt ligand palmitoylation[15]; these data suggest that accumulation of β-catenin in NMR cells was likely independent of autocrine Wnt signaling (Supplementary Fig. S3a). Furthermore, overexpression of NMR Axin1 did not decrease β-catenin abundance (Supplementary Fig. S3b). These results raise the possibility that β-catenin has unique functions that are independent of canonical Wnt/β-catenin signaling in NMRs.

Despite abundant accumulation of β-catenin in NMR cells, expression of cyclin D1, a mitogenic factor and primary downstream target of the Wnt/β-catenin pathway, decreased (Fig. 1a). Due to lower expression of mitogenic factors, NLFs and NSFs grew much more slowly than mouse cells, which exhibited exponential growth (Fig. 1d). These unexpected observations suggest unique alterations in the β-catenin signaling pathway in NMR cells, and that this axis could be relevant to the unique attributes of NMR cells.

**β-catenin depletion induces senescence-like phenotypic changes in NSFs**. To identify the functions of β-catenin in NMR cells, we performed shRNA knockdown of *β-catenin* in NSFs. Immunoblot analysis revealed that β-catenin expression was repressed by *β-catenin* shRNAs, and that expression of cyclin D1 was reduced (Fig. 2a). A TOPFLASH reporter assay also demonstrated that *β-catenin* knockdown attenuated transcriptional activity significantly[16] (Fig. 2b). Phenotypically, *β-catenin* knockdown resulted in noticeable morphological changes in NSFs, such as increased cell size and cell surface area (Fig. 2c). Furthermore, proliferation of *β-catenin*-knockdown NSFs slowed markedly (Fig. 2d), indicating that β-catenin promotes cell proliferation in NSFs. Because the phenotype of *β-catenin*-knockdown NSFs was similar to that of senescent cells, we measured the activity of senescence-associated-β-galactosidase (SA-β-gal). *β-catenin* knockdown induced marked activation of SA-β-gal (Fig. 2e and Supplementary Fig. S4). In addition, *β-catenin* knockdown induced nuclear accumulation of p21, a CDK1 inhibitor used as a marker of cellular senescence (Fig. 2f), and promoted formation of 8-hydroxy-2′-deoxyguanosine (8-OHdG), a biomarker for DNA damaged by oxidative stress[17,18] (Fig. 2g). Induction of SA-β-gal activation and 8-OHdG formation was not affected by treatment with an empty vector (*shControl*) or a vector carrying a non-target shRNA (*shNTControl*), supporting the specific effects of shRNAs targeting *β-catenin* (Supplementary Fig. S5). Taken together, these findings suggested that *β-catenin* knockdown induces phenotypic changes associated with cellular senescence in NSFs.

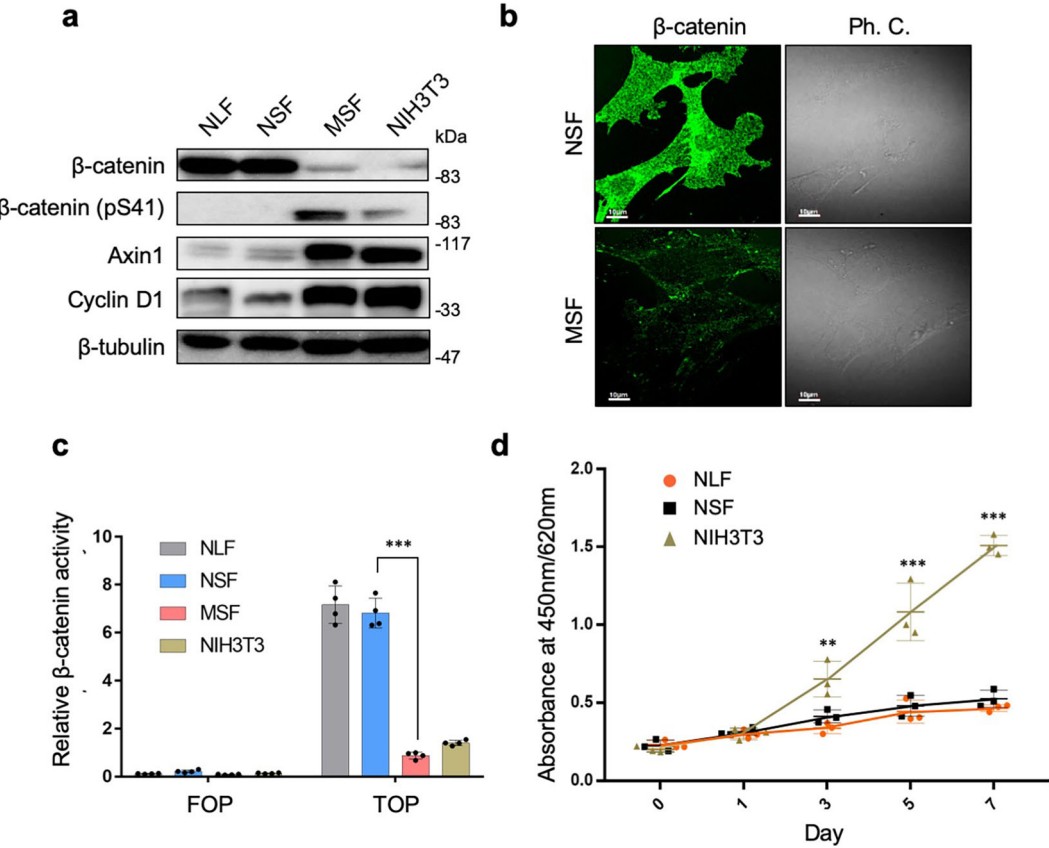

**Fig. 1 Unique activation of β-catenin signaling in NMR cells. a** Representative immunoblots comparing expression of β-catenin and its downstream targets in NMR lung fibroblasts (NLFs), NMR skin fibroblasts (NSFs), mouse fibroblasts (MSFs), and NIH 3T3 cells. **b** Representative images of immunofluorescence staining of β-catenin in NSFs and MSFs. In the phase-contrast images, unknown microbodies are abundant in NSFs. **c** TOPFLASH reporter assay showing significant differences in relative β-catenin activity between NLFs, NSFs, and NIH 3T3 cells. **d** The proliferation rates of NSFs, NLFs, and NIH 3T3 cells were determined in a growth assay conducted for 7 days. Data presented in **c, d** are expressed as the mean ± standard deviation; ***$p <$ 0.001, two-sided Student's $t$ test.

**Accumulation of cholesterol-enriched lipid droplets is associated with β-catenin abundance.** We also observed that NMR cells accumulated abundant microbodies, which were identified as lipid droplets by Oil Red O (ORO) staining using two different solvents: 2-propanol and triethyl-phosphate (TEP) (Supplementary Fig. S6). Quantitative analysis also revealed that NSFs contained more abundant lipid droplets than MSFs under multiple cultured conditions (Supplementary Fig. S7). Because ORO staining with TEP was more sensitive than 2-propanol staining, we used TEP to stain lipid droplets with ORO in subsequent experiments (Fig. 3a). Transmission electron microscopy (TEM) further confirmed that round lipid droplets were more abundant in NSFs than in NIH 3T3 cells (Fig. 3b). Interestingly, we found that lipid droplet formation was abolished by β-catenin knockdown (Fig. 3c and Supplementary Fig. S5c), suggesting a functional link between lipid droplet formation and β-catenin-mediated senescence-like phenotypic changes.

To explore this possibility, we analyzed the contents of NSF lipid droplets. Quantitative analysis of total cholesterol revealed that cholesterol was more abundant in NSFs than in MSFs, and that levels fell significantly after β-catenin knockdown (Fig. 4a). By contrast, levels of triglycerides, another lipid droplet component, were comparable between NSFs and MSFs (Supplementary Fig. S8a). These results suggest that lipid droplets in NSFs comprised primarily cholesterol rather than triglycerides. The contribution of cholesterol to β-catenin-dependent lipid droplet formation was examined further using CholEsteryl

BODIPY FL $C_{12}$, a cholesterol transport tracer. Fluorescence analysis revealed that cholesterol transport into ORO-positive lipid droplets within NSFs was inhibited by β-catenin knockdown (Fig. 4b), implying involvement of β-catenin signaling in regulation of cholesterol transport.

To assess the physiological relevance of cholesterol-enriched lipid droplets in NSFs, we examined the effects of inhibiting cholesterol synthesis on cellular phenotypes. Treatment of NSFs with lovastatin, an HMG-CoA reductase inhibitor, decreased cellular cholesterol content to levels equivalent to those in β-catenin knockdown cells; this effect was dose-dependent (Fig. 4c). Under conditions of cholesterol inhibition, NSFs exhibited SA-β-gal activation, as observed in β-catenin knockdown cells (Fig. 4d, e and Supplementary Fig. S8b). These findings suggest that cholesterol is crucial for protecting NMR cells from senescence-like phenotypic changes.

To further examine if the phenotypic changes in NMR cells induced by β-catenin knockdown were indeed due to cellular senescence, NSFs were treated with the DNA cross-linking agent mitomycin C to simulate therapy-induced senescence. Treatment of NSFs with mitomycin C increased SA-β-gal activity to a level similar to that in β-catenin knockdown NSFs (Supplementary Fig. S9a, b). Mitomycin C also induced accumulation of p21 in the nucleus, although it did not affect lipid droplet formation (Supplementary Fig. S9). Furthermore, reloading cholesterol into β-catenin knockdown senescent-like cells failed to suppress induction of SA-β-gal activity and 8-OHdG formation, indicating

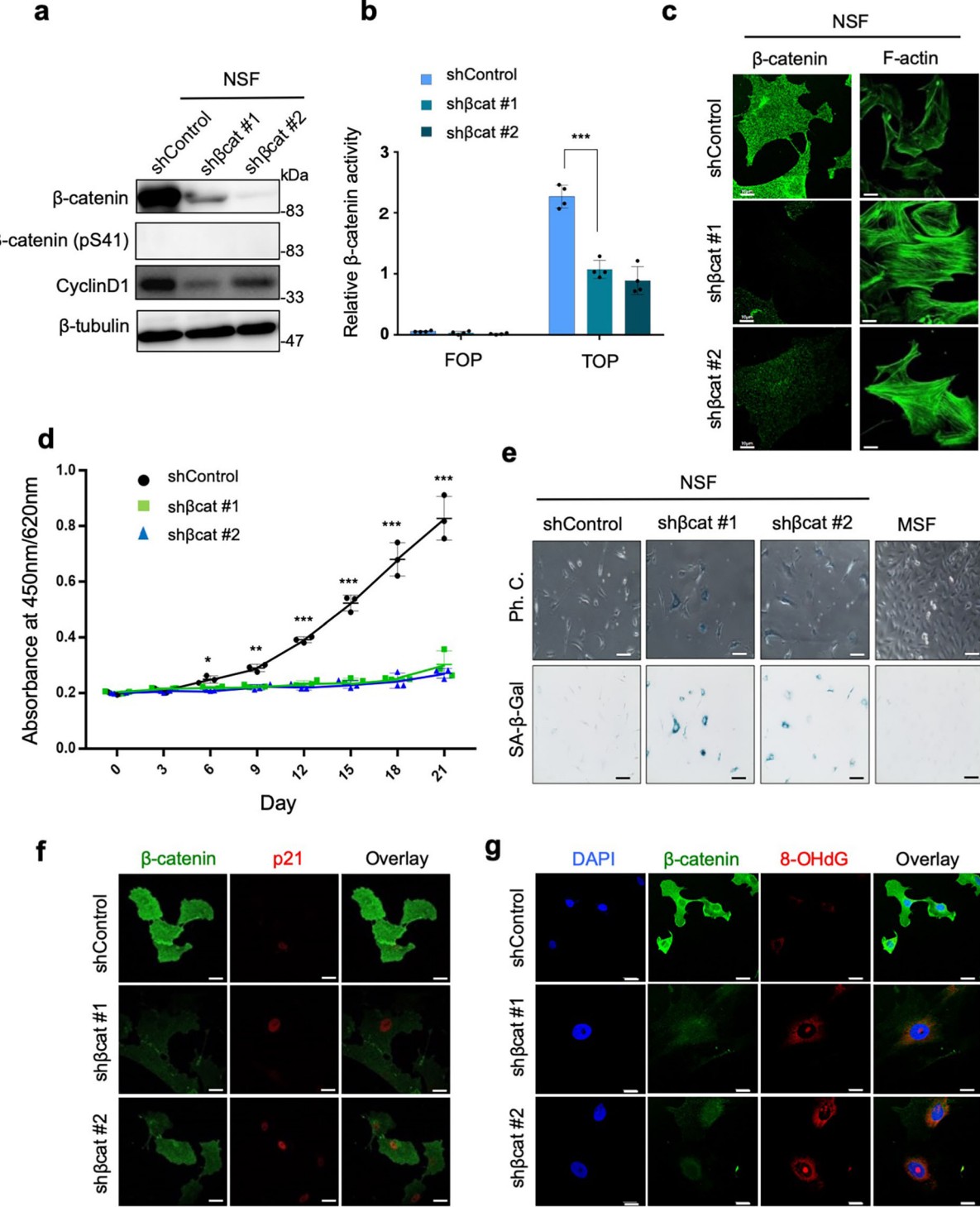

**Fig. 2 β-catenin knockdown induces senescence-like phenotypic changes in NSFs. a** Immunoblots showing changes in expression of components of the β-catenin pathway after *β-catenin* knockdown in NSFs. β-tubulin was used as a loading control. **b** TOPFLASH assay showing relative β-catenin activity of NMR skin fibroblasts (NSFs) after *β-catenin* knockdown. **c** Morphological changes in NSFs caused by *β-catenin* knockdown. Scale bars, 10 μm. **d** Proliferation rates of NSFs and their β-catenin knockdown counterparts were determined in a growth assay. Data presented in b and d are expressed as the mean ± standard deviation (*n* = 4); ***P < 0.001, two-sided Student's *t* test. **e** Representative images showing SA-β-gal activity in control NSFs, *β-catenin* knockdown NSFs, and MSFs (Left). Quantitative analysis of SA-β-Gal activity in control NSFs and *β-catenin* knockdown NSFs (Right). Data are expressed as the mean ± standard deviation (*n* = 4 biologically independent experiments) (lower graph); ***P < 0.001, two-sided Student's *t* test. **f** Immunofluorescence staining of β-catenin (green) and p21 (red) in control NSFs and *β-catenin* knockdown NSFs. **g** Immunofluorescence staining of β-catenin (green) and 8-OHdG (red) in control and *β-catenin* knockdown NSFs. Scale bar, 20 μm.

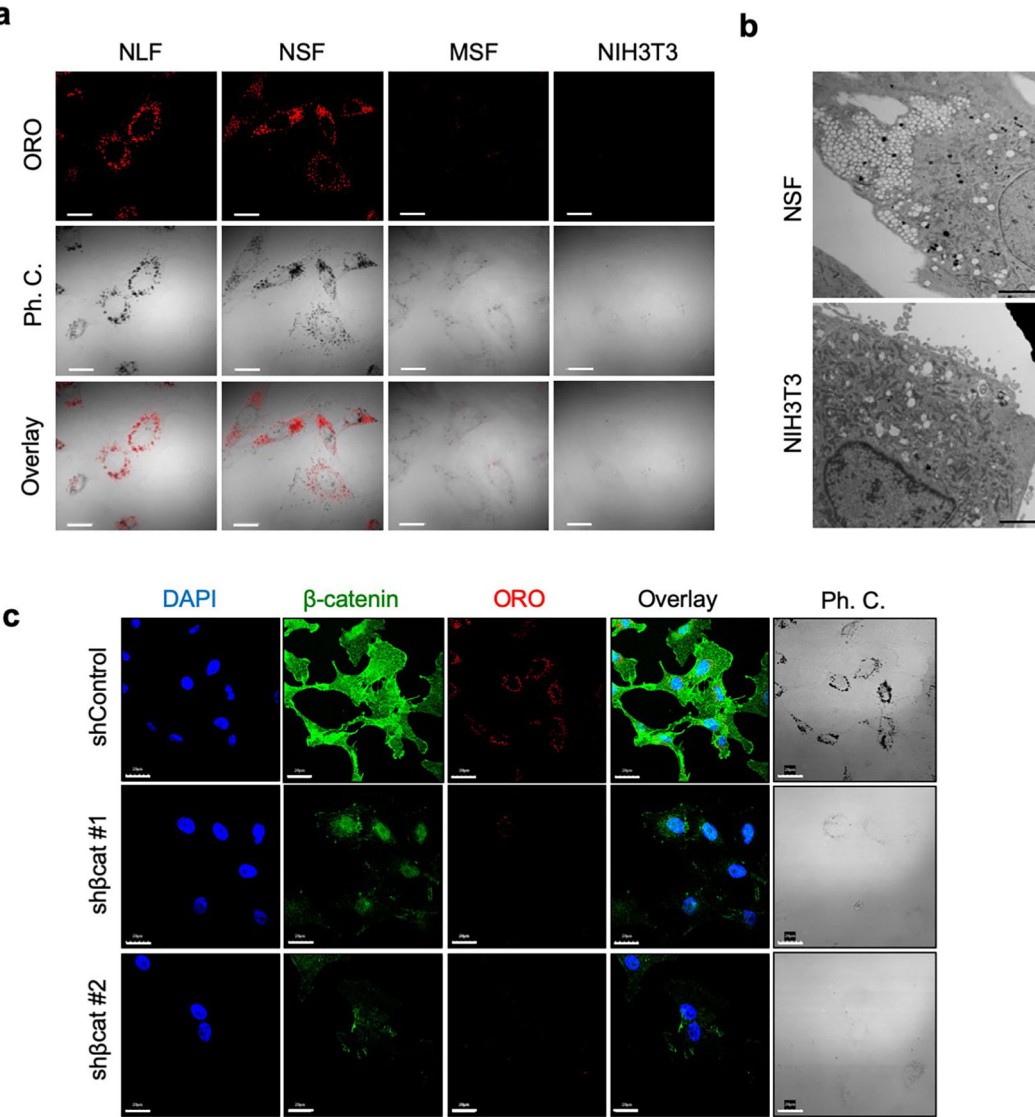

**Fig. 3 Association between β-catenin abundance and lipid droplet accumulation. a** Representative confocal images showing staining of lipid droplets by ORO in NLFs, NSFs, MSFs, and NIH 3T3 cells. Scale bar, 20 μm. **b** Representative TEM images showing robust lipid droplet accumulation in NSFs. Scale bar, 5 μm. **c** Representative fluorescence images of DAPI, β-catenin, and ORO staining, along with phase-contrast images, of control and *β-catenin* knockdown NSFs. Scale bar, 20 μm.

that the cellular events induced by *β-catenin* knockdown were irreversible (Supplementary Fig. S10). These observations demonstrated that the phenotypic changes induced by *β-catenin* knockdown and/or cholesterol depletion were tightly associated with cellular senescence.

As cumulative population doubling could increase SA-β-Gal activity[19,20] and lipid droplets are widely distributed in aged cells[21,22], we performed ORO staining of NSFs at different passage numbers to determine if NMR cells accumulated lipid droplets as replicative senescence progressed. Lipid droplet abundance was unchanged, irrespective of passage number (Supplementary Fig. S11), corroborating that supernumerary lipid droplets in NSFs were a unique feature of NMR cells.

**Relationship between β-catenin abundance and the LXR/RXR pathway in NMR cells.** To determine the mechanisms by which β-catenin induces accumulation of cholesterol-enriched lipid droplets, we performed comparative RNA-seq analysis of

control and *β-catenin* knockdown NSFs. Raw RNA-seq data was submitted under Gene Expression Omnibus (GEO) accession number GSE147871. Ingenuity Pathway Analysis (IPA) revealed that the LXR/RXR pathway, which modulates cholesterol metabolism and lipogenesis, was downregulated significantly by *β-catenin* knockdown (Fig. 5a). Among the genes involved in the LXR/RXR pathway, only apolipoprotein F (ApoF), a secreted glycoprotein that associates with LDL/HDL, was downregulated markedly by *β-catenin* knockdown (Fig. 5b and Supplementary Table S1), which was corroborated by RT-PCR analysis (Fig. 5c). ApoF inhibits cholesteryl ester transfer protein-mediated cholesterol transfer between lipoproteins[23–27] (Fig. 5d). Therefore, upregulation of ApoF in NMRs likely suppresses the transfer of cholesterol among lipoproteins, which could contribute to accumulation of cholesterol within lipid droplets. As a consequence of LXR/RXR pathway suppression by *β-catenin* knockdown, NF-κB was activated and its downstream targets (*Il1B*, *Mmp9*, *Msr1*, and *Ptgs2*) were upregulated (Fig. 5b, e and Supplementary Table S2). These findings suggest that

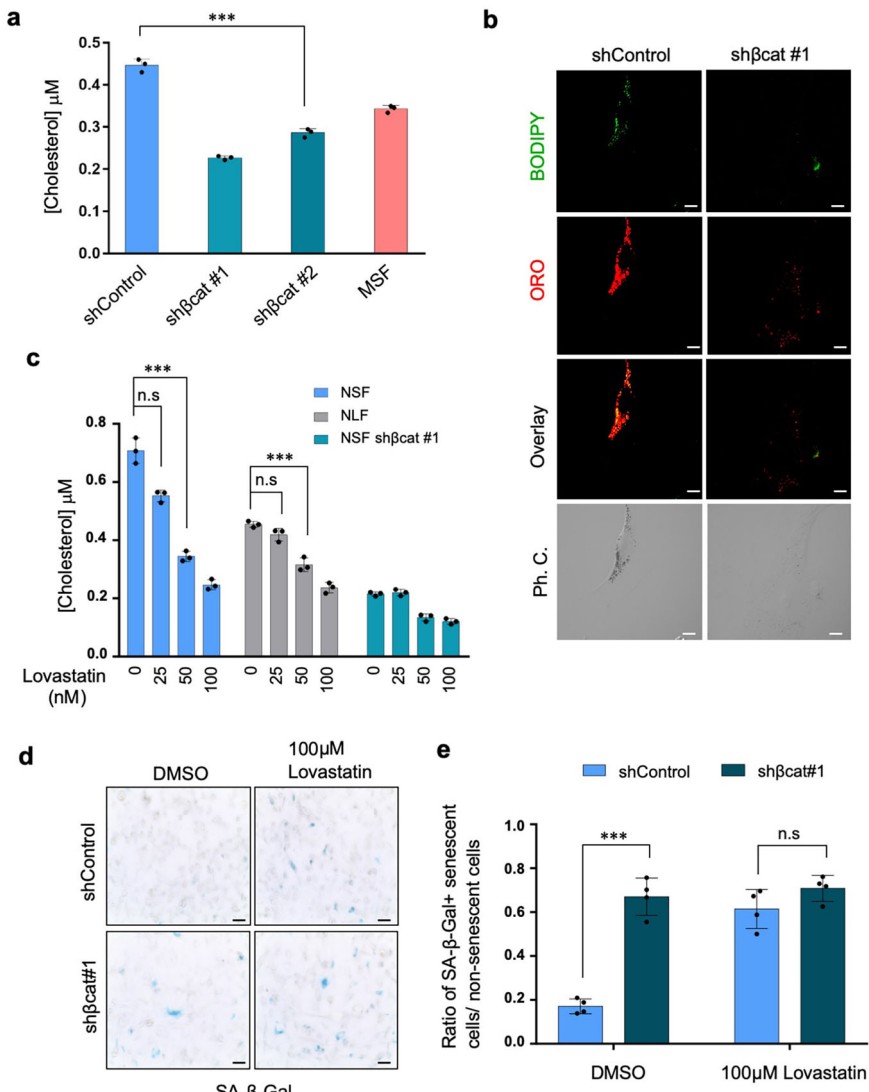

**Fig. 4 Association between senescence-like phenotypes and accumulation of cholesterol-enriched lipid droplets. a** Cholesterol assay demonstrating that *β-catenin* knockdown decreases the cholesterol concentration. **b** Representative confocal images showing staining of cholesteryl ester by BODIPY C12 in NSF *shControl* and *shβ-catenin* cells, with or without serum. Scale bar, 20 μm. **c** The cholesterol concentration was decreased by lovastatin treatment in a dose-dependent manner. Data are expressed as the mean ± standard deviation ($n = 3$ biologically independent experiments); ***$P < 0.001$, two-sided Student's $t$ test. **d** Representative bright-field images showing that inhibition of cholesterol synthesis in NSFs increases SA-β-gal activity. **e** Quantitative analysis of SA-β-Gal-stained cells in control and *β-catenin* knockdown NSFs. Data are expressed as the mean ± ± standard deviation ($n = 4$ biologically independent experiments); ***$P < 0.001$, two-sided Student's $t$ test.

accumulation of lipid droplets in NSFs is attributable to activation of the LXR/RXR pathway via upregulation of ApoF.

**Knockdown of *ApoF* or *β-catenin* has similar effects in NSFs.** To verify the role of ApoF in NMR β-catenin signaling, we genetically ablated *ApoF* in NSFs. Expression of ApoF protein was suppressed by *shApoF* as well as *shβcat* (Fig. 6a, b and Supplementary Fig. S12a). *ApoF* knockdown significantly suppressed ORO-positive lipid droplet formation (Fig. 6c) and promoted SA-β-gal activation (Fig. 6d). These findings indicate that *ApoF* knockdown mimics the effects of β-catenin knockdown in NSFs, suggesting that ApoF has roles downstream of β-catenin signaling.

**ApoF restores lipid drop formation and prevents development of senescence-like phenotypes under *β-catenin* knockdown conditions.** To further elucidate the function of ApoF, we overexpressed *ApoF* in control and *β-catenin* knockdown NSFs

(Supplementary Fig. S12b, c). Remarkably, *ApoF* overexpression restored formation of ORO-positive lipid droplets in *β-catenin* knockdown NSFs, supporting the notion that ApoF is crucial for lipid droplet accumulation (Fig. 7a). Moreover, ApoF overexpression prior to *β-catenin* knockdown prevented *β-catenin* knockdown-induced expansion of the cell surface area and SA-β-gal activation (Fig. 7b). Notably, ApoF overexpression after *β-catenin* knockdown did not restore cell area expansion or prevent SA-β-gal activation in NSFs (Fig. 7b, c), consistent with the earlier observation that reloading cholesterol in senescent-like cells failed to suppress SA-β-gal induction (Supplementary Fig. S10). Furthermore, fluorescence analysis with BODIPY revealed that cholesterol accumulation in ORO-positive lipid droplets was also dependent on the β-catenin-ApoE axis (Supplementary Fig. S13). These findings suggest that ApoF is required for β-catenin-induced formation of cholesterol-enriched lipid droplets, which may protect NMR cells from senescence-like phenotypic changes.

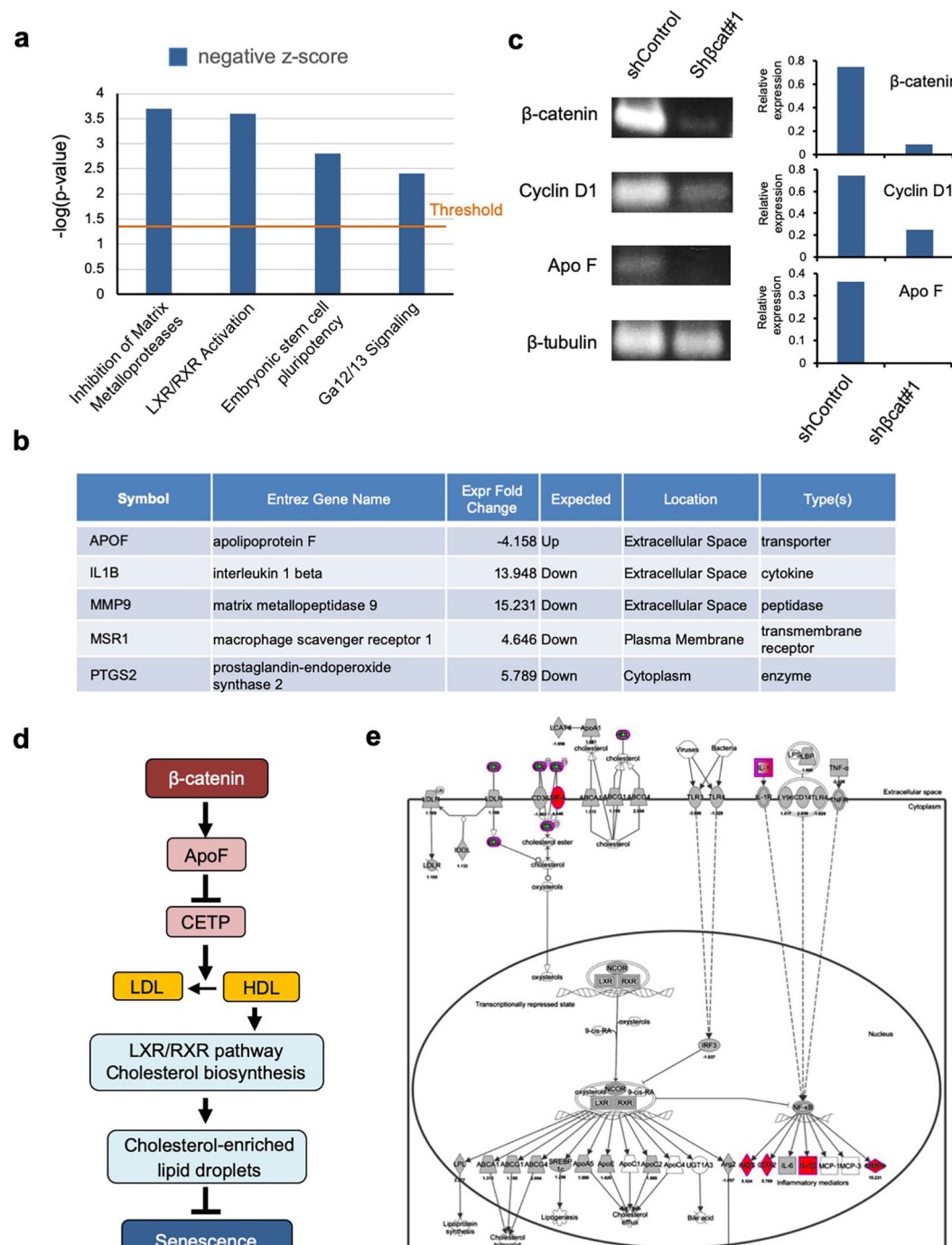

**Fig. 5 Association between β-catenin abundance and LXR/RXR activation in NSFs. a** Ingenuity Pathway Analysis (IPA) results showing pathways that were significantly affected (>4-fold-change) by *β-catenin* knockdown in NSFs. **b** Genes showing significant expression changes in *β-catenin* knockdown conditions. **c** Analysis of target genes regulated by β-catenin in NSFs by RT-PCR (left panels). Quantitative data are also shown (right-most graphs). **d** Schematic diagram illustrating potential links between β-catenin, ApoF, the LXR/RXR pathway, cholesterol biosynthesis and metabolism, lipid droplets, and senescence in NMR cells. **e** Schematic IPA diagrams showing LXR/RXR-dependent changes in gene expression potentiated by *β-catenin* knockdown.

**Protective effects of the β-catenin-ApoF axis against oxidative stress.** Above, we showed that *β-catenin* knockdown increased formation of 8-OHdG, a biomarker of oxidative stress, suggesting that cellular senescence regulated by the β-catenin-ApoF axis is linked to oxidative stress (Fig. 2g and Supplementary Figs. S5c, 10b). To assess this possibility, we examined the effects of *β-catenin* or *ApoF* knockdown on reactive oxygen species (ROS) levels using CellROX Green Reagent, based on a previous

observation that cellular senescence resulted from the accumulation of oxidative damage inflicted by ROS[28]. Evidently, *β-catenin* and *ApoF* knockdown led to a marked increase in ROS levels compared with those in the control cells (Fig. 8a). Next, we used the Alamar Blue Assay to measure the cellular reducing power of NSFs as an index of cell viability. Viability of both *β-catenin* and *ApoF* knockdown NSFs fell significantly (Fig. 8b). However, treatment with N-acetyl-L-cysteine, a strong

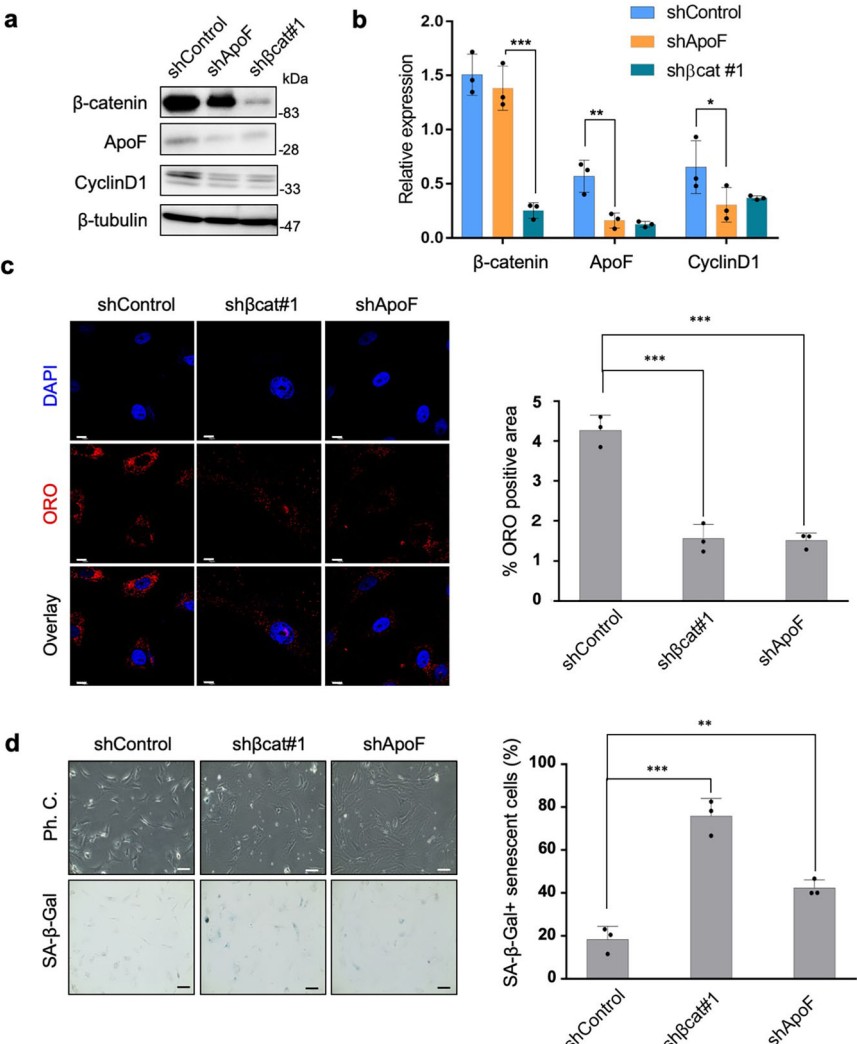

**Fig. 6 Knockdown of *ApoF* or *β-catenin* has similar effects in NSFs. a** Immunoblot confirming downregulation of ApoF upon *ApoF* and *β-catenin* knockdown in NSFs. β-tubulin was used as a loading control. **b** Densitometric quantification of β-catenin and ApoF expression from the immunoblots shown in **a**. Data are expressed as the mean ± standard deviation ($n = 3$ biologically independent experiments). *$P < 0.05$ and ***$P < 0.001$, two-sided Student's paired $t$ test. **c** Immunofluorescence images demonstrating decreased abundance of lipid droplets upon *β-catenin* or *ApoF* knockdown (left). Scale bars, 10 μm. Quantitative results showing percentage coverage of cells stained with ORO versus that by total cells in the ORO assay (right). Data are expressed as the mean ± standard deviation ($n = 3$ biologically independent experiments). ***$P < 0.001$, two-sided Student's $t$ test. **d** Representative images showing SA-β-gal activity in control, *β-catenin*, and *ApoF* knockdown NSFs (left). Quantitative analysis of SA-β-Gal-stained cells in control, *β-catenin*, and *ApoF* knockdown cells (right). Data are expressed as the mean ± standard deviation ($n = 3$ biologically independent experiments); n.s., non-significant; **$P < 0.01$ and ***$P < 0.001$, two-sided Student's paired $t$ test.

antioxidant[29], increased viability of *β-catenin/ApoF* knockdown NSFs. These findings suggest that the β-catenin-ApoF axis protects NMR cells from the oxidative damage that induces cellular senescence. Furthermore, our findings suggest that β-catenin/ApoF-mediated upregulation of cholesterol uptake and subsequent lipid droplet formation suppress NSF senescence.

**Differential role of β-catenin in NMR and mouse cells**. Finally, to investigate whether the interconnection between β-catenin and lipid droplet abundance could be manifested in mice, we over-expressed β-catenin in MSFs and NIH3T3 cells. However, ApoF expression remained unchanged (Supplementary Fig. S14a), and there was no marked accumulation of lipid droplets in either MSFs or NIH3T3 cells (Supplementary Fig. S14b). In addition, we observed that NSFs had no ability for anchorage-independent growth, regardless of abundant β-catenin expression (Supplementary Fig. S15a), while β-catenin overexpression in mouse

cells promoted anchorage-independent growth (Supplementary Fig. S15b), indicating that β-catenin is oncogenic in mouse cells but not in NMR cells. These results further highlight the functional difference between NMR and mouse β-catenin, and suggest that the β-catenin-regulated cholesterol metabolism via ApoF is unique to the NMR cells.

## Discussion
In most normal mammalian cells, activation of Wnt/β-catenin signaling promotes cell cycle progression by upregulating expression of mitogenic proteins such as c-Myc and Cyclin D1[30]. Furthermore, aberrant upregulation of the Wnt/β-catenin pathway is a major cause of colorectal cancer[31,32], which is primarily attributed to mutations in APC, Axin1, APC, or β-catenin[33,34]. Paradoxically, NMR cells expressed high levels of β-catenin and low levels of Axin1, yet target proteins such as cyclin D1 and c-Myc were expressed at considerably lower levels, potentially accounting for

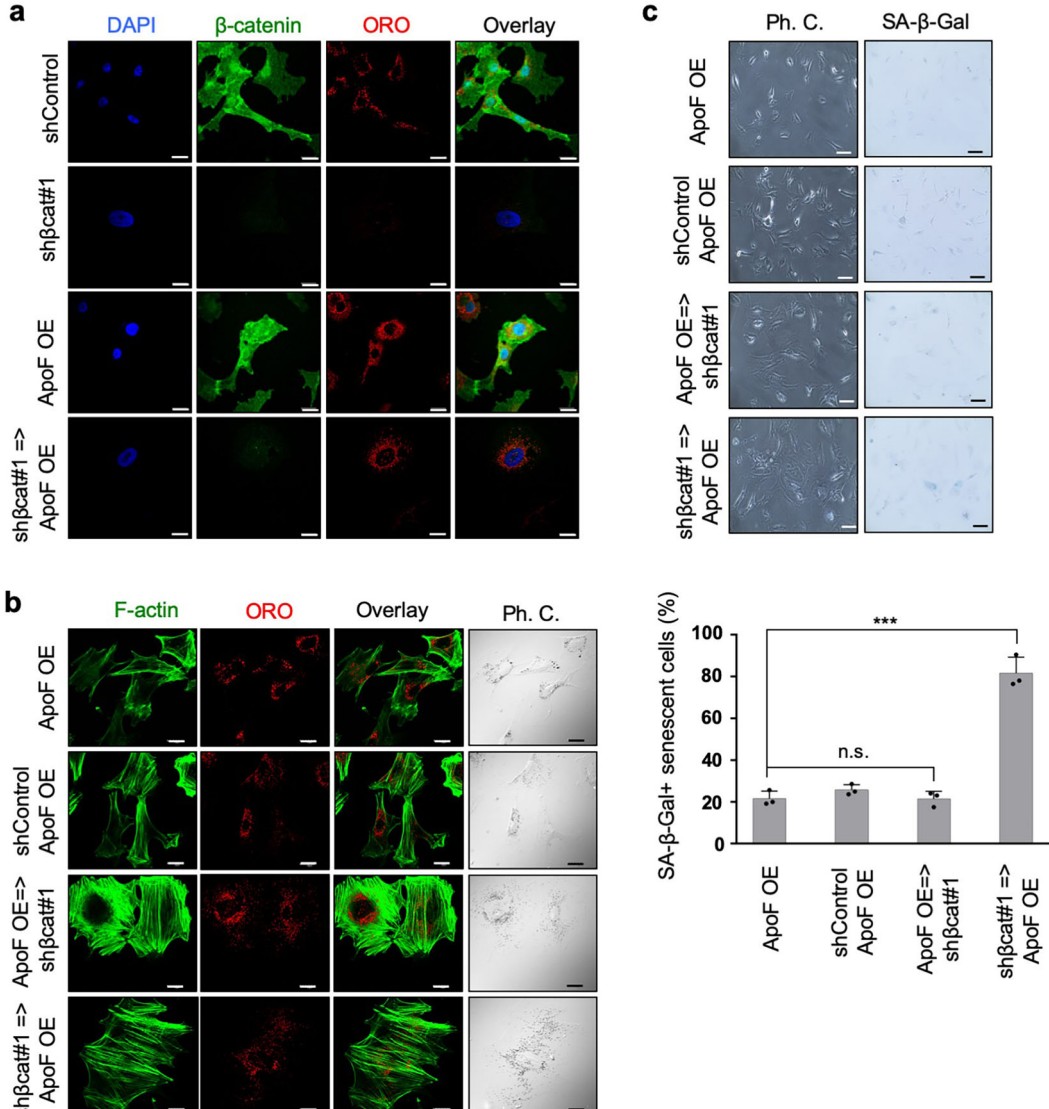

**Fig. 7 ApoF restores lipid droplets under conditions of β-catenin depletion. a** Immunofluorescence images demonstrating lipid droplet restoration by *ApoF* overexpression. Scale bars, 20 μm. **b** Immunofluorescence images demonstrating changes in cell size and actin rearrangement upon *β-catenin* knockdown and/or *ApoF* overexpression. Scale bars, 20 μm. **c** SA-β-gal assay of NSFs expressing the indicated shRNA and/or ApoF cDNA (left). Scale bars, 100 μm. Quantitative results showing the percentage coverage of senescent cells versus that by total cells in the SA-β-gal assay (right). Data are expressed as the mean ± standard deviation (*n* = 3 biological replicates). ***P < 0.001 and n.s., not significant; two-sided Student's *t* test.

the slower growth rate of NMR cells. Nonetheless, transcriptional activity of β-catenin increased in NMR cells. Furthermore, neither inhibition of Wnt production nor Axin1 overexpression affected β-catenin accumulation in NMR cells. These unexpected observations suggest potential unique roles for β-catenin independent of the canonical Wnt/β-catenin pathway in NMRs.

*β-catenin* knockdown in NSFs altered cellular morphology and function, with a considerable expansion of cell area, significantly decreased growth rate, increased SA-β-Gal activity, nuclear accumulation of p21, and increased DNA damage. These phenotypes were consistent with cellular senescence, indicating that *β-catenin* knockdown induced cellular senescence in NSFs. By contrast, *β-catenin* knockdown abolished accumulation of cholesterol-enriched lipid droplets, and inhibition of cholesterol synthesis caused senescence-like phenotypic changes similar to those observed after *β-catenin* knockdown. These observations suggested that abundant β-catenin in NMR cells suppresses cellular senescence via cholesterol accumulation in lipid droplets,

consistent with a study that reported the suppressive role of cholesterol on senescence in mice[35]. We also demonstrated that the β-catenin-ApoF axis was associated with ROS levels and cellular reducing power. In addition, cellular senescence is induced by accumulation of oxidative damage inflicted by ROS of mitochondrial origin[28]. A recent study showed that mitochondrial ROS generation rates are comparable between NMRs and mice, but that the capacity to neutralize ROS is much higher in NMRs than in mice. In conjunction with these findings, our results support a hypothesis that the abundant cholesterol in NMR cells could serve as a ROS scavenger that suppresses onset of cellular senescence. This would be consistent with a previous study demonstrating that cholesterol can function as an antioxidant to counter oxidative stress[36]. However, we could not rule out the possibility that β-catenin could also contribute to additional molecular mechanisms underlying the anti-senescence effects; for example, upregulation of the matrix metalloprotease pathway and GPCR signaling (Fig. 5a).

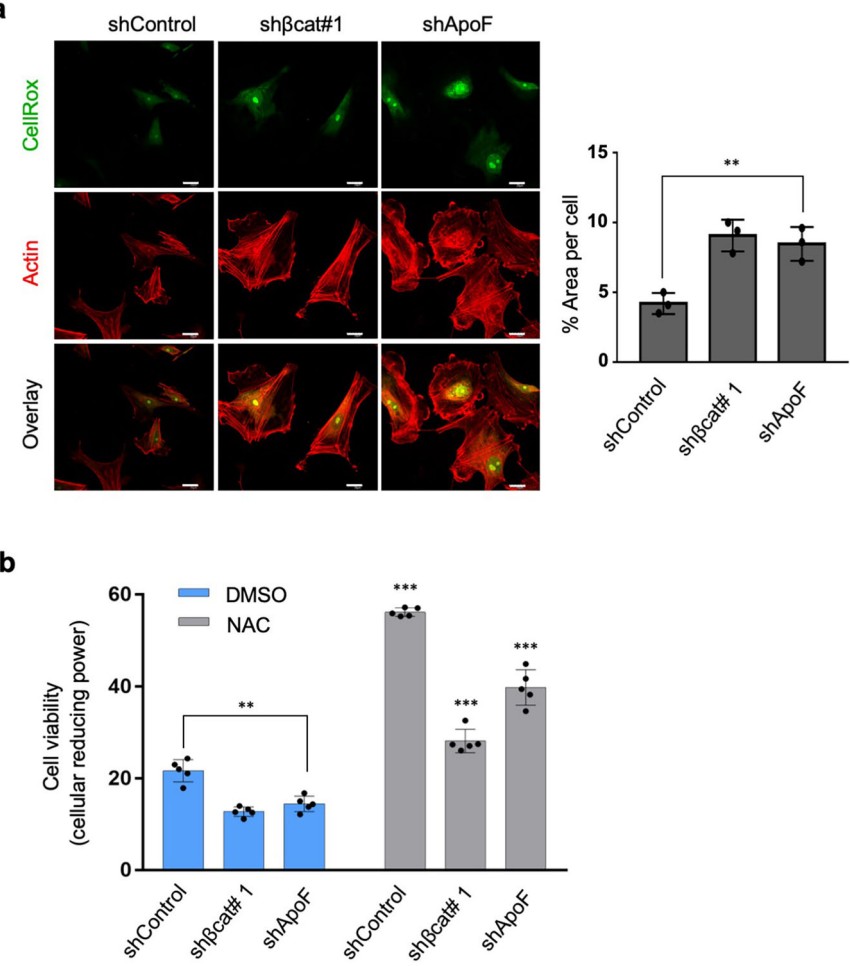

**Fig. 8 Induction of oxidative stress and decreased cell viability under β-catenin and ApoF knockdown conditions. a** Immunofluorescence images showing elevated oxidative stress detected by CellRox upon β-catenin or ApoF knockdown. Scale bars, 20 µm. (left) Quantitative analysis of the CellRox in control, β-catenin, and ApoF knockdown NSFs (right). Data are expressed as the mean ± standard deviation ($n = 3$ biologically independent experiments) (lower graph); **$P < 0.01$, Student's unpaired $t$ test. **b** Alamar Blue assay showing decreased viability of NSFs upon β-catenin or ApoF knockdown, which was restored by treatment with NAC. Data are expressed as the mean ± standard deviation (n = 5 biologically independent experiments). **$P < 0.01$ and ***$P < 0.001$, Student's unpaired $t$ test.

To date, several studies have attempted to identify the functional link between Wnt/β-catenin signaling and lipid droplet formation in other contexts, but the results have been somewhat contradictory. A cancer stem cell model revealed a direct correlation between Wnt pathway activity and increased lipid contents[37]. On the contrary, under K-Ras overexpression conditions, canonical Wnt signaling reprograms lipid metabolism by attenuating lipid droplet accumulation[38]. In breast cancer cells, β-catenin knockdown upregulates proteins associated with lipid metabolism[39]. In the present study, however, we found that β-catenin knockdown in NMR cells suppressed the LXR/RXR pathway involved in cholesterol metabolism and lipogenesis, suggesting that the β-catenin pathway positively regulates cholesterol metabolism in NMRs. These seemingly contradictory roles for β-catenin in regulation of lipid droplet formation suggest that β-catenin exerts context-dependent, and further underscore the unique functions of β-catenin in NMRs.

We identified ApoF, a cholesterol transfer inhibitor protein, as a unique target of β-catenin in NMR cells[40]. ApoF knockdown suppressed lipid droplet accumulation and promoted cellular senescence in a manner similar to β-catenin knockdown, while ApoF overexpression restored lipid droplet formation in β-catenin knockdown NSFs. These results suggest that ApoF is a

crucial mediator of β-catenin-mediated cholesterol accumulation in lipid droplets. However, the mechanisms by which ApoF expression is regulated by β-catenin are yet to be elucidated. We attempted to identify the functional link between TCF4, β-catenin, and ApoF using PNU-74654, an inhibitor that prevents interaction between β-catenin and TCF[41]. We found that treatment with PNU-74654 did not alter ApoF expression (Supplementary Fig. S16), indicating that ApoF expression is TCF-independent. This suggests that a unique transcription factor(s) contributes to the β-catenin-mediated regulation of ApoF expression in NMR cells, or alternatively that ApoF expression is regulated indirectly via an unknown mechanism downstream of the β-catenin signaling.

In summary, we identified constitutively elevated β-catenin activity in NMR cells. This increased β-catenin activity promoted accumulation of cholesterol-enriched lipid droplets via ApoF upregulation, which protected NMR cells from cellular senescence. These findings confirmed that NMR cells are intrinsically susceptible to cellular senescence[42], potentially due to their low rate of basal metabolism[43], which could be beneficial for longevity and cancer resistance. Hence, upregulation of the unique β-catenin pathway in NMR cells could counterbalance its strong senescence potential, thereby promoting longevity and survival

under harsh conditions at the whole-organism level. Further analyses of the molecular mechanisms underlying the anti-senescence functions of cholesterol may reveal unique approaches to treating aging-related conditions.

## Methods

**Antibodies and chemicals.** Alexa Fluor 488-phalloidin, Alexa Fluor 594-goat anti-rabbit IgG, HRP-conjugated goat anti-rabbit IgG, and HRP-conjugated goat anti-mouse IgG were purchased from Thermo Fisher Scientific (Waltham, MA, USA). Anti-β-catenin (D10A8), anti-phospho-β-catenin (pSer41), anti-Axin1 (C76H11), anti-GSK-3β, anti-phospho-GSK-3β (pSer9), anti-cyclin D1, and anti-p21 antibodies were from Cell Signaling Technology (Beverly, MA, USA). The anti-8-OHdG (15A3) antibody was from Santa Cruz Biotechnology (Dallas, TX, USA) and anti-β-tubulin was purchased from Sigma Aldrich (St Louis, MO, USA). The anti-ApoF (ab231585) antibody was from Abcam (Cambridge, MA, USA). The protease inhibitor cocktail was purchased from Nacalai Tesque (Kyoto, Japan). ORO and triethyl-phosphate [$(C_2H_5O)_3PO$] were purchased from WAKO (Osaka, Japan). Lovastatin was purchased from Merk (Darmstadt, Germany).

**Cell culture.** Primary adult NMR fibroblasts were received from the Department of Aging and Longevity Research, Kumamoto University. MSFs were prepared from adult mouse skin, and NIH 3T3 cells were obtained from American Type Culture Collection. To investigate whether temperature, oxygen concentration, or serum concentration affected β-catenin expression or lipid droplet abundance, NMR and mouse cell lines were cultured in Dulbecco's modified Eagle's medium (DMEM) supplemented with 10% FBS or 15% FBS under hypoxic (5.0% $CO_2$, 3.0% $O_2$) or normoxic (5.0% $CO_2$, 16.5% $O_2$) conditions at 32 °C or 37 °C. Upon reaching 70–80% confluency, cells were harvested and lysates prepared. To maintain optimal culture conditions for NMR cell lines, primary NMR fibroblasts and mouse fibroblasts between passage numbers 1–13 were grown in DMEM supplemented with 15% (v/v) fetal bovine serum (FBS) at 32 °C under hypoxic conditions, as described previously[14].

**Inactivation of Wnt signaling.** The Wnt production inhibitor, IWP-2 (Selleck Chemicals), was used to inactivate Wnt signaling in NMR fibroblasts. NMR fibroblasts were plated at 500,000 cells/well in six-well plates. After 48 h, cells were treated with vehicle (0.1% DMSO) or IWP-2 in a dose-dependent manner (up to 100 μM), followed by lysate collection after 48 h. Lysates were then used for immunoblotting. To inhibit the interaction between β-catenin and TCF-4, NMR and MSFs were seeded for 48 h and then exposed to the small-molecule compound PNU-74654 (Selleck Chemicals) at concentrations of 50, 100, and 200 μM.

**Plasmid constructs and shRNA constructs.** To generate *β-catenin* shRNA, sequences were designed to target the central and UTR regions, and the silent constructs were subcloned into the pLKO1 lentiviral vector. *ApoF* and *β-catenin* cDNA was generated by PCR using NMR and mouse cDNA as templates, respectively. The *ApoF* and *β-catenin* constructs were then subcloned into the CSII-CMV-MCS-IRES-Bsd lentiviral vector (RIKEN) and the PCX4 retroviral vector, respectively. All constructs were confirmed by sequencing. The pLKO1 non-targeting shRNA control plasmid was purchased from Sigma Aldrich. The oligo-nucleotide sequences used to generate shRNA vectors in this study are listed in Supplementary Table S3.

**Retroviral and lentiviral infection.** For the knockdown experiment, lentiviral shRNA targeting NMR β-catenin and ApoF, or an empty vector construct, was packaged into HEK293T cells along with a pLKO1 vector containing the packaging genes *Vsv-G*, *Gag-pol*, and *Rev* from the MISSION Lentiviral packaging mix (Sigma Aldrich), using Lipofectamine 3000 as the transfection reagent. The 293T culture medium containing lentiviral particles was collected, passed through a 0.2 μm syringe filter (Whatman), and supplemented with 4 μg/mL Polybrene (Nacalai Tesque). For transduction, cells cultured in a 6 cm plate were incubated with 2 mL filtered virus-containing medium and growth medium (1:1 ratio) overnight. This point was designated as Day 0. The viral medium was then replaced with fresh growth medium on Day 1 post-transduction. On Day 3 post-transduction, trans-duced cells were selected by culture with 10 μg/mL puromycin. For ApoF over-expression in NMR cells, a lentiviral packaging vector (pcAG-HIVgp), a Rev-expressing vector (pCMV-VSV-G-RSV-Rev), and a CSII-CMV-MCS-IRES-Bsd lentiviral vector containing the NMR ApoF genetic construct were transfected into PLT cells using FuGene (Promega, Madison, WI, USA). The culture supernatant was then used as a source of the virus, followed by transduction into NSFs as described previously. For dual lentiviral transfections, the second transduction was conducted on Day 7 post-transduction. For β-catenin overexpression, the retroviral vector PCX4 was used to transduce mouse cell lines. The production and infection of retroviral vectors were performed as described previously[44].

**RNA-sequencing.** Prior to RNA sequencing, puromycin-selected cells were harvested on Day 7 post-transduction and replated onto a 6 cm dish ($1 \times 10^6$ cells).

Medium containing puromycin was changed every other day until the cells reached 80% confluency. Cells were transfected with *shControl* (Empty vector) or *shβ-catenin #1* lentiviral vector, and total RNA was extracted using Sepasol-G (Nacalai Tesque), according to the manufacturer's protocol. Library preparation was per-formed using a TruSeq stranded mRNA sample prep kit (Illumina, San Diego, CA), according to the manufacturer's instructions. Sequencing was performed on an Illumina HiSeq 2500 platform in 75 bp pair-end mode. Illumina Casava1.8.2 software was used for base-calling. Sequenced reads were mapped to the NMR reference genome sequences (GCA_000247695.1) using TopHat v2.0.13 in combination with Bowtie2 ver. 2.2.3 and SAMtools ver. 0.1.19. The fragments per kilobase of exon per million mapped fragments were calculated using Cufflinks version 2.2.1. For further analysis, the 216 genes showing a greater than fourfold change in expression were detected using IPA[45].

**Immunoblotting.** Cells were lysed in RIPA buffer (20 mM Tris-HCl pH 7.4, 150 mM NaCl, 0.5 mM EDTA, 1% NP40, 1 mM PMSF, 1 mM sodium orthovanadate, 50 mM NaF, and sodium deoxycholate) in combination with a protease inhibitor (Invitrogen), and protein concentration was determined using a Bradford protein assay (BioRad). Ten micrograms of denatured lysates were subjected to 10% SDS-PAGE, and blotted using standard procedures. For protein detection, blots were incubated overnight with primary antibodies, followed by a secondary antibody (rabbit-HRP; G.E. Healthcare) for 30 min. Chemiluminescence was used to visualize protein bands (G.E. Healthcare).

**Oil-Red O staining.** Cells (25–50% confluency) were seeded onto tissue culture chamber slides and allowed to grow for 48 h before staining. Cells were washed briefly with PBS, followed by fixation for 1 h in 3.7% formaldehyde. Excess for-maldehyde was removed by three brief rinses in deionized water. Prior to staining, 35 mg of ORO were dissolved in 10 mL of 2-propanol or 6 mL TEP and used as a stock solution[46]. A working solution of ORO containing 6 mL of ORO stock solution and 4 mL of deionized water was prepared, followed by filtering (0.2 μm). Subsequently, slides were immersed for 5 min in the working solution of ORO.

**Immunofluorescence.** Cells (25–50% confluency) were seeded onto tissue culture chamber slides and allowed to grow for 24–36 h before fixation (manipulation). For experiments involving tracking of cholesterol ester and visualization of oxidative stress in cells, cells were incubated in medium containing BODIPY CholEster $C_{12}$ (Thermo Fisher Scientific) and CellRox Green (Thermo Fisher Scientific) for 2 h prior to fixation. Cells were rinsed in PBS (10 mM phosphate, pH 7.5/100 mM NaCl) and subsequently fixed with 4% paraformaldehyde in PBS for 10 min at room temperature. Following PBS rinsing, cells were permeabilized with 0.1% Triton X-100 in PBS (TPBS) for 10 min at room temperature. Cells were then incubated with 2% BSA for 1 h at room temperature. Cells were rinsed with TPBS and incubated at 4 °C overnight with primary antibody diluted in antibody buffer (Blocking One). On the following day, cells were rinsed three times with TPBS before incubation with a secondary antibody conjugated to fluorescein (Alexa-Fluor) for 30 min to 1 h at room temperature. Cells were then rinsed three times with TPBS and mounted with ProLong Gold (Molecular Probes) for immuno-fluorescence microscopy. For combined immunofluorescence and ORO staining, after three exchanges of PBS following application of the appropriate fluorescein-conjugated fluorescein, glass slides were immersed in the ORO working solution for 5 min. Slides were rinsed three times with deionized water, followed by mounting with ProLong Gold.

**TEM.** Cells were cultured on a polystyrene coverslip (Cell Desk; Sumitomo Bakelite Co., Ltd., Japan), fixed with 2% formaldehyde and 2.5% glutaraldehyde in 0.1 M sodium-phosphate buffer (pH 7.4), and washed for 5 min (three times) in the same buffer. Cells were post-fixed for 1 h with 1% osmium tetroxide and 1% potassium ferrocyanide in 0.1 M sodium-phosphate buffer (pH 7.4), dehydrated in a graded series of ethanol solutions, and embedded in Epon812 (TAAB Co. Ltd., U.K.). 80 nm ultra-thin sections were stained with saturated uranyl acetate and lead citrate solution. Electron micrographs were obtained under a JEM-1400plus transmission electron microscope (JEOL, Japan).

**TOPFLASH reporter assay.** Cells were seeded into 24-well plates (in triplicate) at a density of $5 \times 10^4$ cells/well in a total volume of 500 μL complete growth medium. On the subsequent day, the cell lines were transfected with reporter vectors (ratio of FOPFLASH/TOPFLASH-Firefly luciferase:pRL-TK-Renilla luciferase = 10:1) using Lipofectamine 3000 (Invitrogen), in which the pRL-TK reporter vector was used as an internal control. 24 h after transfection, luciferase activity was measured using a PicaGene Dual Sea Pansy Luminescence Kit (Wako).

**Cell proliferation assay.** Cells were plated on a 96-well dish at a density of 500 cells/well in a total volume of 100 μL of complete growth medium. After the cells had fully attached to the surface of the dish, 10 μL of Cell Counting Kit WST-8 (Doujin Chemistry Laboratories) reagent was added to each well, followed by a 1-h incubation at 32 °C. Cell density was determined using a microplate reader

(absorbance 450 nm). On the subsequent days, cell density was measured from 7 to 20 days of culture, and growth rates were plotted.

**SA-β-Gal assay**. Cells were seeded at $2 \times 10^4$ cells/well in a 12-well plate 48 h prior to staining. Cells were washed briefly with cold PBS before fixation in 1 mL of 0.5% glutaraldehyde, followed by incubation at 4 °C for 5 min. Cells were stained with 2 mL of freshly prepared 5-bromo-4-chloro-3-indoyl-β-D-galactopyranoside (X-Gal) staining solution, followed by incubation at 37 °C for 5 h. Staining was terminated by washing three times with ice-cold PBS for 5 min. Color images of X-Gal-stained cells were captured with bright-field settings, mounted under an inverted light microscope, and imaged using a 10× objective lens. Stained cells were counted and analyzed manually. Image J analysis was utilized to confirm consistency. Chlorophenol red β-D-galactopyranoside was also used for quantitative analysis of SA-β-Gal activity.

**RT-PCR**. RNA was extracted from cells with Sepasol-RNA I Super G (Nacalai Tesque), followed by reverse transcription using ReverTra Ace qPCR RT Master Mix (TOYOBO) to obtain cDNA. PCR was then performed, and the intensity of DNA bands stained by SYBR gold was quantified using ImageJ analysis. The nucleotide sequences of the primers used are listed in Supplementary Table S4.

**Cholesterol assay**. Cells were seeded at $1 \times 10^5$ cells/well in a six-well plate 48 h prior to cholesterol extraction. Cells were washed briefly with cold PBS (twice), followed by addition of 1 mL of hexane:isopropanol (3:2) to the wells for lipid extraction, followed by incubation at room temperature for 30 min. The lipid-containing mixture was recovered in an Eppendorf tube and air-dried using an Iwaki Halogen Vacuum Concentrator (IVC-500) for 20 min at room temperature. The pellet was then resuspended and cholesterol content was determined using the Amplex Red™ Cholesterol Assay Kit (Invitrogen). For reloading cholesterol, cholesterol-methyl-β-cyclodextrin (C4951, Sigma) was added to the culture medium at the final concentration of 10 μg/mL for 24 h[47,48]. Incorporation of cholesterol into cells was estimated by adding CholEsteryl BODIPY FL C$_{12}$ (Thermo Fisher Scientific) to the culture medium (concentration, 5 μM) for 2 h before fixation and immunofluorescence analysis.

**Alamar Blue assay**. Cultures ($1 \times 10^4$ cells/well in a 24-well plate) were set up in complete medium (DMEM supplemented with 15% (v/v) FBS, penicillin/streptomycin, 2 mM L-glutamine (Nakalai Tesque) plus 0.1 mM non-essential amino acids (Nakalai Tesque). Cultures were incubated for 48 h, followed by replacement with fresh medium containing AlamarBlue Reagent (Thermo Fisher Scientific) (medium:AlamarBlue ratio = 9:1). After 12 h of incubation with Alamar Blue, the medium was collected and colorimetrically measured at absorbances of 570 nm (Oxidized form of AlamarBlue Reagent) and 600 nm (Reduced form). The percentage reduction in cell number or cell viability was calculated using the formula stated in the manufacturer protocol.

**Soft agar colony formation assay**. A soft agar colony formation assay was performed as described[49]. Briefly, single-cell suspensions ($1 \times 10^4$ cells) in 1.5 ml of DMEM containing 15% FBS and 0.36% agar were plated in 12-well culture dishes containing a layer of 2.5 ml of the same medium containing 0.7% agar. At 14 days after plating, colonies were stained with 3-(4,5-dimethylthiazol-2-yl)-2,5-diphenyltetrazolium bromide, micrographs obtained, and cell numbers counted.

**Statistics and reproducibility**. All results are presented as the mean ± standard deviation of at least three biological replicates, as indicated in the figure legends. All data analyses were conducted using GraphPad Prism 7. Unpaired two-tailed $t$ tests or one-way or two-way ANOVA were used to determine $P$ values. $P$ values of *<0.05, **<0.01, and ***<0.001 were deemed significant.

**Reporting summary**. Further information on research design is available in the Nature Research Reporting Summary linked to this article.

## Data availability
Raw RNA-seq data with accession number GSE147871 will be released publicly under Gene Expression Omnibus (GEO).

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

## Acknowledgements

The authors acknowledge the NGS core facility of the Genome Information Research Center at the Research Institute for Microbial Diseases of Osaka University for support with RNA sequencing and data analysis, and Dr. Hiroko Ohmori from the Core Instrumentation Facility, Research Institute for Microbial Diseases, for conducting TEM observations of cells. Raw RNA-seq data related to this study were submitted under GEO accession number GSE147871. This work was supported by JPSP KAKENHI (Grant numbers 19H03504 and 19H04962 to M.O.; 19H03412 to T.I.; and 18H02365 to K.M.) and AMED (JP19gm5010001 to T.I., and JP19gm5010001 and 19gm0704040 to K.M.).

## Author contributions

W-Y.C. and Y.K. conceptualize the ideas of investigating and analyzing the β-catenin-regulated pathway in naked mole-rat (NMR) cells. W-Y.C. designed and performed the experiments and carried out data analysis. D.O. contributed to the RNA-seq analysis. W-Y.C., J.K., and M.O. interpret the results. K.M. conceived the ideas of contrasting multiple protocols to validate the intrinsicality of NMR. T.I. conceived the idea of extending the analyses of Wnt/β-catenin pathway. W-Y.C., Y.K., and D.O. generated figures. K.K., S.N., and M.O. oversaw the experimental design and analysis. W-Y.C. wrote the original manuscript. M.O., K.M., T.I., and H.O revised the manuscript.

## Competing interests

The authors declare no competing interests.
