## [Peer Review File · Communications Biology]

Reviewers' comments:

Reviewer #1 (Remarks to the Author):

1. Brief summary of the manuscript

The authors report a unique mechanisms of cholesterol metabolism in cells from the long-lived, cancer-resistant Naked mole-rat. This mechanism could be responsible the anti-senescent properties found in NMR fibroblasts. Increased β -catenin activity in these cells is linked to increased accumulation of cholesterol-enriched lipid droplets. Further, β -catenin knockdown or inhibition of cholesterol synthesis abolished lipid droplet formation and enhanced induction of senescence-like phenotypes. The group pursued this observation further by looking at β -catenin-regulated genes. They discovered that ablation of the upregulated apolipoprotein F and the LXR/RXR pathway, suppressed lipid droplet accumulation and promoted cellular senescence. This suggested that apolipoprotein F is a critical mediator of the β -catenin signaling in NMR cells. This could be a strategy used by the naked mole-rat to promote longevity in NMRs.

2. Overall impression of the work

The work is pretty thorough and well-written. The manuscript is easy to follow and flows logically. This is definitely a significant work both for technical innovation of the manipulations of these primary cells, and to get to a more mechanistic explanation of the myriad of NMR age-defying phenotypes.

3. Specific comments, with recommendations for addressing each comment

I recommend this manuscript for publication with a minor revision or clarification here and there. There are a few points that could improve the experience for a reader, but do not hinder the overall story.

a. This statement leaves the reviewer hanging, "Identification of transcription factors that associates with β -catenin would shed light on the unique functions of β -catenin in NMRs." (pg 9 end of paragraph 1) Could the authors speculate which non-canonical transcription cotters might be involved?

b. In the methods section, I would like further clarification on the sample sizes used. The authors state 3 independent experiments. How many cell lines? How many biological replicates? Are the sample sizes that they refer to, biological or technical replicates?

c. It is worth reminding the reader in the figure legends what NSF/NLF, etc stand for. I found it in the body of the text "NMR skin/lung fibroblasts (NSFs/NLFs), mouse skin fibroblasts (MSFs), and NIH 3T3 cells" but good to have in other locations.

d. I don't know how helpful figure 5d is. That could be in supplemental with figure S10 brought to the main part of the manuscript since it is in effect a summary of the conclusion.

Reviewer #2 (Remarks to the Author):

In Chee et al. 2020, the authors show compelling evidence that b-catenin signaling in naked mole rat

cells is altered compared to mouse cells, and they propose an ApoF-driven cholesterol metabolism phenotype downstream of b-catenin knockdown as a contributing mechanism. Conceptually, the knowledge gap addressed, and the authors' experimental model are well written and well supported by the literature. However, lack of methodological information, issues with data presentation, and lack of critical mechanistic experiments in mouse cells all confuse interpretation or sometimes even reviewer assessment of the experimental model presented.

Major comments:

1. The writing, literature, and structure of the introduction and discussion are clear and concise, and they are generally well done.
2. The authors propose that b-catenin signaling is altered in naked mole rat cells compared to mouse cells, through a mechanism involving an ApoF-dependent switch in cholesterol metabolism. However, the authors do not test critical components of the mechanism in mouse cells and do not clarify experimentally—or (alternatively) sufficiently discuss conceptually—how this mechanism is relevant to differences in senescence or cancer between species.
3. The comparison between NMR cells and mouse cells is complicated by several variables that are either unclear as presented or not addressed. The culturing methods are incompletely reported, and the culture conditions appear different between NMR and mouse cells. Do conditions (confluence, media composition, temperature, oxygen, etc.) affect b-catenin expression or activity? How similar are the b-catenin pathway member protein sequences between species, and could differences in protein levels be explained by differences in antibody affinity? Do NIH-3T3 cells, which are transformed, have normal Wnt pathway signaling?
4. Presentation of the data is sometimes difficult for the reader to interpret. Overexposure in Western blots of loading controls and/or b-catenin is present throughout the paper. Lighter exposures should be used for loading controls where possible, and multiple exposures should be shown when necessary—particularly Figs. 1, 6, S1, and S9. Many of the phase contrast or brightfield images do not have enough contrast to interpret.
5. The analysis of the RNAseq in Fig. 5 is opaque. Going through all of the provided documents multiple times, I can find no information in the methods used, no raw data, nor alternatively any reference to a publication or database. The text and figure do not mention number of replicates, criteria for statistical significance, parameters for input gene lists used in IPA analysis, parameters used to create the gene list in Fig. 5b, etc. The rationale for choosing ApoF as a primary target is therefore unclear—was it found through unbiased profiling of b-catenin-dependent gene expression changes? Were b-catenin target genes significantly decreased upon knockdown? What was the timing of the knockdown? Expansion of the RNA-seq analysis could potentially—at least provisionally—answer many of the questions posed in the discussion.
6. In addition to the RNA-seq, other methods are incomplete, and the number of replicates and timing of a given experiment is often unclear or not defined. For example, how was the cholesterol measurement conducted?
7. The mechanism connecting b-catenin to ApoF is unclear. The authors show large changes in ApoF RNA levels in response to b-catenin knockdown, and conclude from Fig. S9 that ApoF expression is TCF-independent, suggesting that ApoF expression may be only indirectly related to b-catenin

knockdown. What was the timing of the b-catenin knockdown experiments, and how does the timing influence your interpretation? Is ApoF expression decreased with both b-catenin hairpins? Is there biochemical or bioinformatic evidence linking b-catenin to ApoF expression, for example predicted transcription factor binding sites on the promoter?

Minor comments:

8. The original rationale for measuring species-specific b-catenin levels and activity is unclear.

9. The control for shRNA experiments is unclear; for example, was it a non-targeting hairpin or an empty vector control?

10. The experiment(s) in Fig. 7b and c provide critical mechanistic insight, but are missing empty vector and shControl conditions. The timing of the experiment is not explained, which could influence interpretation. Figure 7c images are labeled incorrectly.

11. The authors' interpretation of Fig. S1a seems incorrect, although it is difficult to tell based on the western blot exposures presented.

12. The manuscript claims antioxidant effects of cholesterol multiple times. Does cholesterol treatment rescue 8-oxoguanine induction in b-catenin knockdown cells?

13. Claims of lipid droplet composition are made without evidence.

14. Bar charts are shown instead of individual data points.

15. The statistical comparisons seem to be drawn on the figure incorrectly on Fig. 6b.

Point-by-point response to the referees' comments

We thank the editor and referees for time spent carefully reviewing our manuscript. We have responded to the comments below, with the referees' comments in **black** and the authors' responses in **blue**. We have made three significant changes to the manuscript, none of which were requested by the referees. We have included experiments involving CellRox to evaluate the oxidative status of NSF's under control, *β-catenin*, and *ApoF* knockdown conditions, and further validated oxidative stress using an Alamar Assay (Pg 8 Line 15-30) (Fig. 8). We also included results from soft agar colony formation assays showing anchorage-independent growth of wild-type NSF's and MSF's (pg 9 Line 3-7) (Supplementary Fig. S14).

Reviewer #1

1. Brief summary of the manuscript

The authors report unique mechanisms of cholesterol metabolism in cells from the long-lived, cancer-resistant naked mole-rat. This mechanism could be responsible for the anti-senescent properties found in NMR fibroblasts. Increased β -catenin activity in these cells is linked to increased accumulation of cholesterol-enriched lipid droplets. Further, β -catenin knockdown or inhibition of cholesterol synthesis abolished lipid droplet formation and enhanced the induction of senescence-like phenotypes. The group pursued this observation further by looking at β -catenin-regulated genes. They discovered that the ablation of the upregulated apolipoprotein F and the LXR/RXR pathway suppressed lipid droplet accumulation and promoted cellular senescence. This suggested that Apolipoprotein F is a critical mediator of the β -catenin signalling in NMR cells. This could be a strategy used by the naked mole-rat to promote longevity in NMRs.

2. Overall impression of the work

The work is pretty thorough and well-written. The manuscript is easy to follow and flows logically. This is definitely a significant work both for technical innovation of the manipulations of these primary cells and to get to a more mechanistic explanation of the myriad of NMR age-defying phenotypes.

3. Specific comments, with recommendations for addressing each comment

I recommend this manuscript for publication with a minor revision or clarification here and there. There are a few points that could improve the experience for a reader but do not hinder the overall story.

Reviewer #1 (Remarks to the Author):

a. This statement leaves the reviewer hanging, "Identification of transcription factors that associates with β -catenin would shed light on the unique functions of β -catenin in NMRs." (pg 9 end of paragraph 1) Could the authors speculate which non-canonical transcription coters might be involved?

We agree with the reviewer that further elaboration on this point (i.e., by speculating about whether analysis of non-canonical transcription factors would clarify the mechanism underlying β -catenin-mediated control of cholesterol metabolism). However, we believe that expanding our dataset will be time-consuming, especially in light of the current pandemic, and is beyond the scope of the current manuscript. For this reason, we chose not to speculate in the absence of compelling evidence. Further studies will be conducted to fully elucidate the regulatory mechanisms underlying β -catenin-mediated control of cholesterol metabolism (pg 10, line 31-34).

b. In the methods section, I would like further clarification on the sample sizes used. The authors state three independent experiments. How many cell lines? How many biological replicates? Are the sample sizes that they refer to biological or technical replicates?

We thank the reviewer for pointing this out. We used at least three biological replicates (culture wells). The number of biological replicates is stated for all experiments that required application of statistical tests.

c. It is worth reminding the reader in the figure legends what NSF/NLF, etc. stand for. I found it in the body of the text "NMR skin/lung fibroblasts (NSFs/NLFs), mouse skin fibroblasts (MSFs), and NIH 3T3 cells" but good to have in other locations.

We have updated the figure legends so that readers can conveniently refer to these abbreviations (legend to Fig. 1a).

d. I don't know how helpful figure 5d is. That could be supplemental with figure S10 brought to the main part of the manuscript since it is, in effect a summary of the conclusion.

Given that the pivotal gene, ApoF, is involved in cholesterol metabolism governed by β -catenin, we do agree that the simplicity of the pathways in the schematic diagram would be more relatable for this experiment. Thus, we have revised Figure 5 and moved the previous Supplementary Fig. S10 to Fig. 5 (incorporated as panel d).

Reviewer #2 (Remarks to the Author):

In Chee et al. 2020, the authors show compelling evidence that b-catenin signaling in naked mole-rat cells is altered compared to mouse cells, and they propose an ApoF-driven cholesterol metabolism phenotype downstream of b-catenin knockdown as a contributing mechanism. Conceptually, the knowledge gap addressed and the authors' experimental model are well written and well supported by the literature. However, lack of methodological information, issues with data presentation, and lack of critical mechanistic experiments in mouse cells all confuse interpretation or sometimes even reviewer assessment of the experimental model presented.

Major comments:

1. The writing, literature, and structure of the introduction and discussion are clear and concise, and they are generally well done.
2. The authors propose that b-catenin signaling is altered in naked mole-rat cells compared to mouse cells through a mechanism involving an ApoF-dependent switch in cholesterol metabolism. However, the authors do not test critical components of the mechanism in mouse cells and do not clarify experimentally—or (alternatively) sufficiently discuss conceptually—how this mechanism is relevant to differences in senescence or cancer between species.

We appreciate the reviewer's insightful suggestions and agree that it would be informative to determine if β -catenin controls lipid metabolism in murine cells. We overexpressed β -catenin in mouse cell lines using the retrovirus PCX4 to determine if levels of β -catenin expression similar to those in NMRs would increase ApoF expression and/or affect lipid droplet abundance. Immunoblotting revealed that expression of β -catenin and ApoF were linked (Supplemental Fig. S13a). Representative images of Oil-Red O-stained cells are included as Supplemental Fig. S13b.

To determine if β -catenin is an oncogenic factor in the context of the present study, we also conducted soft agar colony formation assays, as anchorage-independent cell growth is a hallmark of cancer. We performed these analyses using NMR skin fibroblasts and mouse skin fibroblasts, and compared the results with those from control and *β -catenin*-overexpressing murine cells (Supplemental Fig. S14). The findings suggested that NMRs and mice do not share the same pathway, as excess β -catenin did not promote tumorigenicity, but rather protected NMR cells against senescence via cholesterol metabolism. By contrast, ectopic expression of *β -catenin* caused tumorigenicity in mouse cell lines, but did not change ApoF expression or lipid droplet abundance (Supplemental Fig. S13). We have added these results to revised manuscript (pg 8, Line 32-34, pg 9 Line 1-9) .

3. The comparison between NMR cells and mouse cells is complicated by several variables that are either unclear as presented or not addressed. The culturing methods are incompletely reported, and the culture conditions appear different between NMR and mouse cells. Do conditions (confluence, media composition, temperature, oxygen, etc.) affect β -catenin expression or activity? How similar are the β -catenin pathway member protein sequences between species, and could differences in antibody affinity explain differences in protein levels? Do NIH-3T3 cells, which are transformed, have normal Wnt pathway signaling?

We apologize for this oversight. We agree with the reviewer that further elaboration of this point and inclusion of new data would be helpful. Thus, we performed the experiment using two cell lines, NSF and MSF, under different conditions: FBS concentration, 10% or 15%; oxygen concentration, hypoxic (3.0%) or normoxic (16.5%); and temperature, 32°C or 37°C (pg11 Line 28-33). The primary purpose of this experiment was to investigate whether β -catenin expression and lipid droplet abundance were affected by the culture conditions. We then compared β -catenin expression between NMR and mouse cells, and found that expression of β -catenin in NMR cells was significantly higher than that in mouse cells, and that lipid droplet abundance remained unchanged under different culture conditions (pg 4, Line 10-14 and Supplemental Fig. S2).

However, we did not perform the Dual-Luciferase Reporter (DLR) assay to test β -catenin activity, as the DLR assay only shows = trans localization of β -catenin to the nucleus, followed by its binding to the transfected TOPFLASH vector. Besides, the pivotal phenotype of NSFs, i.e., lipid droplet abundance, remained unchanged under all culture conditions.

To address the protein sequence similarity of β -catenin among species, we compared NMR, mouse, and human β -catenin sequences. The sequence was 99% conserved among the species, with only one amino acid residue difference (Ser206 in NMR instead of Asp206 in human and mouse; pg 4 line 7-9 and Supplemental Fig. S1a). The antibody used (CST anti- β -catenin (D10A8 XP)) targeted an epitope within the residues surrounding Pro714 of the human β -catenin sequence, which is conserved in both NMR and mice (species reactivity of CST anti- β -catenin (D10A8 XP): human, mouse, rat, and monkey). Therefore, there are no differences in antibody affinity (pg 4, line 7-10).

In NIH3T3 cells, we confirmed that expression of β -catenin and β -catenin was similar to that in MSFs using immunoblot analysis and DLR assays (Fig. 1). We also determined if *β -catenin* overexpression promoted anchorage-independent growth in a soft agar colony formation assay (Supplemental Fig. S14), corroborating the finding that Wnt signaling functions normally in NIH3T3 cells.

4. Presentation of the data is sometimes difficult for the reader to interpret. Overexposure in Western blots of loading controls and/or β -catenin is present throughout the paper. Lighter exposures should be used for loading controls where possible, and multiple exposures should be shown when necessary—particularly Figs. 1, 6, S1, and S9. Many of the phase contrast or brightfield images do not have enough difference to interpret.

We apologize for this oversight and have updated the figures with ones in which the loading control has lower exposure. Regarding the phase-contrast and brightfield images, we have enlarged and cropped the original photos to provide a clearer view and to aid interpretation.

5. The analysis of the RNAseq in Fig. 5 is opaque. Going through all of the provided documents multiple times, I can find no information in the methods used, no raw data, nor alternatively any reference to a publication or database. The text and figure do not mention number of replicates, criteria for statistical significance, parameters for input gene lists used in IPA analysis, parameters used to create the gene list in Fig. 5b, etc. The rationale for choosing ApoF as a primary target is therefore unclear—was it found through unbiased profiling of b-catenin-dependent gene expression changes? Were b-catenin target genes significantly decreased upon knockdown? What was the timing of the knockdown? Expansion of the RNA-seq analysis could potentially—at least provisionally—answer many of the questions posed in the discussion.

We apologize that the methodology for the RNA-seq was incomplete in the original submission. We have added the experimental procedures for RNA-seq to the methodology section (pg 13 Line 9-22), which stipulates publication or database, and the parameters for the input gene lists used in IPA. For raw data, the following secure token was created to allow review of records (GSE147871) while retaining private status: [svgjmwkzativdet](https://www.ncbi.nlm.nih.gov/geo/query/acc.cgi?acc=GSE147871). To review GEO accession GSE147871, go to <https://www.ncbi.nlm.nih.gov/geo/query/acc.cgi?acc=GSE147871> and enter the password [svgjmwkzativdet](https://www.ncbi.nlm.nih.gov/geo/query/acc.cgi?acc=GSE147871) (pg 17 line 14-15).

The rationale for selecting ApoF was through an unbiased screening of affected genes using IPA, in which the genes with more than 4-fold changes in expression were categorized into four pathways, as illustrated in Figure 5a. We also added Supplemental Table S1 and S2, which list DEGs upon β -catenin downregulation in NMR cells. Supplemental Table S1 corroborates the significant knockdown of *β -catenin*. Besides that, we conducted RT-PCR to confirm that expression of *β -catenin* and its canonical target gene, *cyclin D1*, was reduced (Fig. 5c).

Regarding the timing of β -catenin knockdown, we have added the missing description of the procedures (see “Retroviral and lentiviral transfection” pg 12, line 23 - pg 13, line 7, and “RNA-seq” pg 13 line 9 – 22, sections).

6. In addition to the RNA-seq, other methods are incomplete, and the number of replicates and timing of a given experiment is often unclear or not defined. For example, how was the cholesterol measurement conducted?

We apologize for this oversight, and have updated the missing methodology (pg 16 Line 9-20). At least three biological replicates (culture wells) were used. The number of biological replicates is now stated for each experiment that required application of statistical tests. The cholesterol concentration in NMR cells was measured using the Amplex Red™ Cholesterol Assay Kit (Invitrogen). We have included a "Cholesterol assay" section in the revised manuscript (pg 16, line 9-20).

7. The mechanism connecting b-catenin to ApoF is unclear. The authors show large changes in ApoF RNA levels in response to b-catenin knockdown and conclude from Fig. S9 that ApoF expression is TCF-independent, suggesting that ApoF expression may be only indirectly related to b-catenin knockdown. What was the timing of the b-catenin knockdown experiments, and how does the timing influence your interpretation? Is ApoF expression decreased with both b-catenin hairpins? Is there biochemical or bioinformatic evidence linking b-catenin to ApoF expression, for example, predicted transcription factor binding sites on the promoter?

We have added a more detailed description of the timing of β -catenin knockdown in the methods section (pg 12 Line 23-33). We have also updated and included immunoblots showing decreased ApoF expression in cells treated with *sh β -catenin* hairpins (Supplemental Fig. S11a).

Unfortunately, we do not currently have biochemical and/or bioinformatic evidence linking β -catenin to ApoF expression. We agree that further elaboration of this point, i.e., by identifying specific transcription factors, would help to elucidate the regulatory mechanisms underlying β -catenin-controlled ApoF expression. However, biochemical identification of transcription factors has not yet been successful, and since the genomic database of the NMR is incomplete, it is difficult to obtain reliable bioinformatic evidence. Therefore, this important question will be addressed in future studies. We have also carefully addressed this point and acknowledged the current gaps in knowledge (see the revised Discussion, pg 10 line 22-34).

Minor comments:

8. The original rationale for measuring species-specific β -catenin levels and activity is unclear.

We apologize that this was unclear. Regarding β -catenin levels, as shown in Supplemental Figure S1a, the β -catenin sequence is highly conserved among NMR, mice, and humans, and the antibody used recognizes the conserved region. Therefore, species-specific β -catenin levels could be compared by immunoblot analysis. Basal β -catenin activity was assessed using a TOPFLASH assay, based on the knowledge that the TCF binding site is also conserved in NMR β -catenin. The TOPFLASH assay highlighted that β -catenin is localized to the nucleus and activates TCF, and corroborated that β -catenin binds to the TCF binding site and that it was expressed in a constitutively active form in NMR cells. We have added a brief explanation of this to the revised manuscript (pg 4 Line 18-21).

9. The control for shRNA experiments is unclear; for example, was it a non-targeting hairpin or an empty vector control?

For most of the shRNA experiments, the empty vector pLKO1 was used as the *shControl*. Due to lack of genomic information for the NMR, we were concerned that scrambled shRNA could induce a stress response in NMR cells, as it could potentially target unintended mRNAs. By contrast, an empty vector control that contains no shRNA insert was used to control for the potential effects of transduction. However, we demonstrated that a non-targeting shRNA from Sigma had no significant effects on NMR cells. Thus, we have included experiments comparing β -catenin expression, lipid droplet abundance, and SA- β -Gal staining in empty vector control (*shControl*) and non-target control (*shNTControl*) cells (Supplemental Fig. S4, pg 5, line 15-20).

10. The experiment(s) in Fig. 7b and c provide critical mechanistic insight but are missing empty vector and *shControl* conditions. The timing of the experiment is not explained, which could influence interpretation. Figure 7c images are labeled incorrectly.

We have updated the figures by showing overexpression of ApoF in *shControl* (Empty Vector) (Fig. 7b). The timing of the experiments is described in the “Retro and lentiviral infection” subsection of the Methods. (pg 12 Line 23-34 & pg 13 Line 1-7) We appreciate the reviewers pointing out this mistake, which has been corrected.

11. The authors' interpretation of Fig. S1a seems incorrect, although it is difficult to tell based on the western blot exposures presented.

We apologize that the original Figure S1a was not informative. We have repeated the experiments using a Wnt inhibitor as an additional control and hope that the new results (Supplemental Fig. S1b) show well-defined changes in β -catenin expression.

12. The manuscript claims the antioxidant effects of cholesterol multiple times. Does cholesterol treatment rescue 8-oxoguanine induction in b-catenin knockdown cells?

We included new experiments to support the antioxidant effects of cholesterol. Cholesterol treatment suppressed induction of 8-OHdG but failed to rescue oxidative damage under *β-catenin* knockdown conditions because irreversible cellular senescence had already occurred in these cells (Supplemental Fig. S9, pg line 25-27). We have also added new data showing the interconnection between the *β-catenin*-ApoF axis and oxidative stress (Fig. 8, pg 8, line 15-30). However, because we did not have direct evidence for the antioxidant effects of cholesterol, we have just discussed this possibility in the revised manuscript (pg 10, Line 2-9).

13. Claims of lipid droplet composition are made without evidence.

We realized that Oil-Red O stains neutral lipids, and recognize that this limitation should be overcome. Hence, we used Cholesteryl Ester BODIPY (Invitrogen) (pg 6 Line 6-10) to verify the localization of cholesteryl esters to lipid droplets (Fig. 4b and Supplemental Fig. S12).

14. Bar charts are shown instead of individual data points.

We have fixed the errors and made changes to every bar graph.

15. The statistical comparisons seem to be drawn on the figure incorrectly on Fig. 6b.

We apologize for this oversight> we have corrected the error by changing the significance sign *** to **.

REVIEWERS' COMMENTS:

Reviewer #1 (Remarks to the Author):

Thank you for your comments in the rebuttal. I am satisfied with the changes that were made. I think in fact that your group went above and beyond the changes that were asked for and this has made for a better manuscript.

Reviewer #2 (Remarks to the Author):

I appreciate the efforts that the authors took to address my comments on the manuscript. These concerns have been thoroughly addressed, especially regarding rationale and methodology. The additional experiments and controls are well done and strongly support the manuscript.

Minor comments:

The manuscript and the rebuttal document mention only one difference in b-catenin amino acid sequence at residue 206 between mouse and NMR. However, there is a second residue different between mouse and NMR at residue 706 (proline (NMR); alanine (mouse)) according to figure S1a and NP_001290232.1 vs. NP_001159374.1. This residue is likely in the epitope used to raise the antibody (region around P714). It is unlikely that this difference has a major effect on antibody affinity, but it is possible. Do you have data from a second antibody to confirm your result? Alternatively, please briefly mention this caveat in the manuscript.

Although I agree that the main message of the manuscript is not altered regarding the new data on cell culture conditions, the cell culture conditions tested do have interesting, moderate effects on b-catenin levels (and to a lesser extent lipid staining) in both species. If these apparent differences are statistically significant, please add some brief discussion of these results.

COMMSBIO-20-2084

Point-by-point response to the referees' comments

We thank the editor and referees for time spent carefully reviewing our manuscript. We have responded to the comments below, with the referees' comments in **black** and the authors' responses in **blue**.

Reviewer #1 (Remarks to the Author):

Thank you for your comments in the rebuttal. I am satisfied with the changes that were made. I think in fact that your group went above and beyond the changes that were asked for and this has made for a better manuscript.

We thank the reviewer for the compliment.

Reviewer #2 (Remarks to the Author):

I appreciate the efforts that the authors took to address my comments on the manuscript. These concerns have been thoroughly addressed, especially regarding rationale and methodology. The additional experiments and controls are well done and strongly support the manuscript.

We thank the reviewer for constructive advice to improve the experiment and support our hypothesis in the research.

Minor comments:

The manuscript and the rebuttal document mention only one difference in b-catenin amino acid sequence at residue 206 between mouse and NMR. However, there is a second residue different between mouse and NMR at residue 706 (proline (NMR); alanine (mouse)) according to figure S1a and NP_001290232.1 vs. NP_001159374.1. This residue is likely in the epitope used to raise the antibody (region around P714). It is unlikely that this difference has a major effect on antibody affinity, but it is possible. Do you have data from a second antibody to confirm your result? Alternatively, please briefly mention this caveat in the manuscript.

Thank you for noticing this. Regarding the antibody, we used anti- β -catenin antibody (D10A8; CST) raised against a synthetic peptide corresponding to residues surrounding Pro714 of human β -catenin protein. This antibody is widely used in the related field and is guaranteed to cross-react with human, mouse, rat, monkey, even though mouse β -catenin has an alanine at residue 706. This suggests that the substitution at the residue 706 does not affect the reactivity of this antibody. By contrast, NMR β -catenin has 100% sequence identity to the residues surrounding Pro714 of human β -catenin, so we used this antibody to detect NMR β -catenin and showed abundant β -catenin expression in NMR cells in this study. We also confirmed the upregulation of β -catenin mRNA by RT-PCR. However, to avoid misleading, we have added a brief explanation for cross-reactivity of the antibody in the text (pg 4, line 10).

“ an antibody cross-reactive with multiple species and ...”

Although I agree that the main message of the manuscript is not altered regarding the new data on cell culture conditions, the cell culture conditions tested do have interesting, moderate effects on b-catenin levels (and to a lesser extent lipid staining) in both species. If these apparent differences are statistically significant, please add some brief discussion of these results.

Thank for pointing out these potentially interesting phenomena. We agree that there are appreciable fluctuations in β -catenin levels under different culture conditions. Particularly, β -catenin levels are

relatively low at high temperature and hypoxia conditions. Although we still do not know the actual reason, it is possible that reduced expression of β -catenin may be due to reduced viability of NMR cells under these conditions. We hypothesize that the metabolism of NMR cells is increased due to elevated temperature, while oxygen levels in the hypoxic condition could not meet the demands of increased metabolism in NMR cells. Therefore, heat stress and hypoxia could act synergistically to induce cellular stresses, resulting in the reduction of cell viability of NMR cells. We have added some comments about the reduction of β -catenin levels at high temperature and hypoxia conditions (pg 4, line 13-16).

“The results showed that β -catenin expression levels in NSF β s were considerably higher than those in MSF β s under any conditions, although they were relatively low at high temperature and hypoxia conditions, potentially due to low cell viability of NMR cells under stressful conditions.”